# Effects of Short-Chain Fatty Acids on Human Oral Epithelial Cells and the Potential Impact on Periodontal Disease: A Systematic Review of In Vitro Studies

**DOI:** 10.3390/ijms21144895

**Published:** 2020-07-11

**Authors:** Gabriel Leonardo Magrin, Franz Josef Strauss, Cesar Augusto Magalhães Benfatti, Lucianne Cople Maia, Reinhard Gruber

**Affiliations:** 1Department of Oral Biology, Medical University of Vienna, Sensengasse 2a, 1090 Vienna, Austria; gabriel.magrin@posgrad.ufsc.br (G.L.M.); drstrauss@odontologia.uchile.cl (F.J.S.); 2Department of Dentistry, Center for Education and Research on Dental Implants, Federal University of Santa Catarina, Campus Reitor João David Ferreira Lima s/n, Florianopolis 88040-900, Brazil; cesarbenfatti@yahoo.com; 3Department of Conservative Dentistry, Faculty of Dentistry, University of Chile, Av. Sergio Livingstone 943, Santiago 7500566, Chile; 4Department of Pediatric Dentistry and Orthodontics, Federal University of Rio de Janeiro, Rua Prof. Rodolpho Paulo Rocco 325, Rio de Janeiro 21941-617, Brazil; rorefa@terra.com.br

**Keywords:** short-chain fatty acids, butyrate, epithelial cells, periodontal disease, periodontitis, in vitro, systematic review

## Abstract

Short-chain fatty acids (SCFA), bacterial metabolites released from dental biofilm, are supposed to target the oral epithelium. There is, however, no consensus on how SCFA affect the oral epithelial cells. The objective of the present study was to systematically review the available in vitro evidence of the impact of SCFA on human oral epithelial cells in the context of periodontal disease. A comprehensive electronic search using five databases along with a grey literature search was performed. In vitro studies that evaluated the effects of SCFA on human oral epithelial cells were eligible for inclusion. Risk of bias was assessed by the University of Bristol’s tool for assessing risk of bias in cell culture studies. Certainty in cumulative evidence was evaluated using GRADE criteria (grading of recommendations assessment, development, and evaluation). Of 3591 records identified, 10 were eligible for inclusion. A meta-analysis was not possible due to the heterogeneity between the studies. The risk of bias across the studies was considered “serious” due to the presence of methodological biases. Despite these limitations, this review showed that SCFA negatively affect the viability of oral epithelial cells by activating a series of cellular events that includes apoptosis, autophagy, and pyroptosis. SCFA impair the integrity and presumably the transmigration of leucocytes through the epithelial layer by changing junctional and adhesion protein expression, respectively. SCFA also affect the expression of chemokines and cytokines in oral epithelial cells. Future research needs to identify the underlying signaling cascades and to translate the in vitro findings into preclinical models.

## 1. Introduction

Periodontitis is characterized by a complex inflammatory response triggered by the presence of a dental biofilm. This bacterial biofilm is able to elicit a dysbiosis in the subgingival microbiome leading to the destruction of the periodontal supporting tissues and eventually tooth loss [1]. Metabolites released by periodontopathic bacteria are capable of provoking an immune response inducing the influx of neutrophils and macrophages to the gingival crevice and epithelium [2,3]. An excessive migration of leukocytes facilitates the periodontal breakdown and is often associated with the initiation and progression of periodontal diseases. Therefore, it can be assumed that the release of secondary metabolites may play a crucial role in periodontal disease [4]. However, the specific role of these metabolites remains unclear.

Short-chain fatty acids (SCFA) are metabolites primarily produced by gastrointestinal bacteria but can also be found in periodontal pockets [5]. They are saturated aliphatic organic acids with one to six carbon molecules. The most common SCFA are acetate (C2), propionate (C3), and butyrate (C4), accounting for over 85% of the total SCFA produced [6]. SCFA derived from gut microbes upon fermentation of fibers, counteract systemic inflammatory and metabolic diseases such as diabetic nephropathy; for example, mice deficient in the metabolite-sensing G protein-coupled receptors GPR43 or GPR109A were not protected by SCFA [7]. On the other hand, SCFA are considered virulence factors when produced locally in periodontal pockets by *Porphyromonas gingivalis, Treponema denticola, Aggregatibacter actinomycetemcomitans, Prevotella intermedia,* and *Fusobacterium nucleatum* [8]. Thus, there is a controversial view on the overall role of SCFA on periodontal health and disease, particularly considering their local production in periodontal pockets [5].

Short-chain fatty acids can regulate the inflammatory response and may represent a link between the microbiota and the immune system [9]. For example, acetate can suppress the accumulation of body fat and liver lipids and, when absorbed in the colon, provokes an increase in cholesterol synthesis [10,11]. Propionate improves insulin sensitivity in the tissues, which is beneficial for obese or individuals with type-2 diabetes, and counteracts cholesterol synthesis thereby decreasing the likelihood for cardiovascular disease [12]. Butyrate can inhibit cell growth of colon carcinomas, suggesting an involvement in the prevention of colon cancer [13,14,15]. In addition, gut-derived SCFA can affect gingival inflammation [16,17,18]. A recent clinical study reported that an anti-inflammatory diet may decrease periodontal parameters due to fibers intake and SCFA production [19].

Apart from serving as main energy source for colonocytes [20], SCFA can affect the differentiation, recruitment, and activation of immune cells by means of cytokines [21]. In endothelial cells, SCFA can modulate the levels of interleukin (IL)-6 and IL-8 [22]. In addition, SCFA can regulate T cell polarization [23], neutrophils recruitment [24], and affect the immunoregulation of monocytes and macrophages [25]. Short-chain fatty acids can regulate the production of IL-18 thereby supporting the integrity of intestinal epithelial cells [26]. The intravenous administration of SCFA is considered a promising therapy in colorectal cancer [27], bowel disease [28], obesity-associated insulin resistance [29], and cardiovascular diseases [30]. However, the extent to which SCFA are involved in health and disease remains to be fully elucidated.

In periodontal research, early in vitro studies suggested that butyrate and propionate have detrimental effects in the periodontium [31]. High concentrations of butyrate induce apoptosis in inflamed human gingival fibroblasts and are associated with periodontal destruction [32,33]. Oral epithelial cells prevent the penetration of bacterial toxins by forming a tight barrier attached to the tooth surface, the so called junctional epithelium [34]. This anatomical structure is not only a passive physical obstacle to microorganisms, but also a natural defense mechanism against pathogenic bacteria [35]. Given that the integrity and the defense mechanisms of epithelial barriers are at least partially controlled by bacteria-released metabolites such as SCFA, it becomes relevant to investigate how SCFA target oral epithelial cells in order to elucidate the possible implications in health and disease.

A recent narrative review proposed a more holistic view on the significance of the oral microbiome that can be either detrimental (pathogenic) or beneficial (commensal) to the host [9]. According to that review, butyrate may influence the oral microbiome composition, favoring periodontal pathogen growth, and stimulating oxidative stress. Considering that butyrate is a vital metabolite to both the oral microbiome and the host, there is thus a demand for a systematic approach to synthetize the current evidence on SCFA in periodontology. The aim of the present study was, therefore, to systematically review the effect of SCFA on human oral epithelial cells and their possible molecular and cellular implications in periodontal disease.

## 2. Results

### 2.1. Selection of Studies and Screening Process

In phase one, 3591 records were identified from five databases. After merging the records and removing duplicates, 2258 records remained. No studies were added from the grey literature since the identified records were within the other databases. Subsequently, 35 references were selected for phase 2. No additional studies were selected from hand search of the reference list of the included studies. After three attempts to contact the experts, no extra data were obtained. A total of 25 studies were excluded in phase 2 (reasons for exclusion can be seen in Appendix A). Finally, 10 articles were qualified for inclusion and data extraction. A flowchart detailing the study selection is available in Figure 1.

### 2.2. Characteristics of the Studies

The included studies were published between 1997 and 2020, and all of them were in English language. They were conducted in five different countries: Austria [36], China [37], Finland [38], Japan [33,39,40,41,42], and the United States of America [43,44]. Concerning the cell lines used, three used primary gingival epithelial cells [36,37,38], six used human oral squamous cell carcinoma cell lines [33,36,39,40,41,42] and three used immortalized transformed human oral epithelial cells [38,43,44]. The SCFA employed in the included studies were butyrate [33,36,37,38,39,40,41,42,44] (C4, 88.1 g/mol), propionate [36,38,43,44] (C3, 74.1 g/mol), acetate [36,43,44] (C2, 60.0 g/mol), lactate [43,44] (not a true fatty acid, C3H6O3, 90.1 g/mol), and formate [43] (C1, 46.0 g/mol). Butyrate or their synonyms (sodium butyrate, butyric acid) were the most used SCFA, applied as intervention in 9 of the 10 included studies [33,36,37,38,39,40,41,42,44]. A summary of the characteristics of the included studies is shown in Table 1.

### 2.3. Risk of Bias in Individual Studies

Using an adapted tool of the World Cancer Research Fund/University of Bristol for cell line studies [45], one study was classified as low risk of bias [37], seven studies as moderate risk of bias [33,36,38,39,40,41,42] and two studies as high risk of bias [43,44]. The most frequent bias was the absence of primary cells from periodontal epithelium in seven studies [33,39,40,41,42,43,44]. Moreover, comparisons between different cell lines were judged “not applicable” in seven studies because only one cell line was used in those investigations [33,37,39,40,42,43,44]. In another two studies, cell lines were not compared in most of the experiments [36,38]. More information about risk of bias assessment is provided in Table 2.

### 2.4. Synthesis of Results

Different methods and assays were applied to withdrawn results in the included studies. Investigations also varied in SCFA used, concentration applied, and treatment regimes, which prevented a direct comparison between the studies, thereby precluding a meta-analysis. Therefore, a narrative synthesis of results was carried out. To facilitate the readability and interpretation of results, the studies were clustered into four groups based on their main results: cell viability, cell morphology changes, cell death mechanisms, and the regulation of adhesion and other molecules. A summary of the data extracted from each article is shown in Table 1. In addition, a schematic flowchart of the potential mechanisms by which SCFA elicit their effects on oral epithelial cells in a periodontal context is shown in Figure 2.

Cell viability: Cytotoxic effects of SCFA on human oral epithelial cells were evaluated by MTT [36,41,43], trypan blue [40,43] and Sytox Green cell death [33,42] assays. Another study applied a cell proliferation test using ^3^H-thymidine incorporation [44]. All studies showed that SCFA reduced cell viability in a dose dependent manner, being butyrate the sole fatty acid in 6 studies [33,37,39,40,41,42]. For example, butyrate induced cell death at around 0.5 mM after 24 h [33] and 48 h incubation [42] with Sytox Green assay. With the MTT assay, 2.5 mM of butyrate for 24 h reduced cell viability by approximately 10% in HSC-3 cells, 20% in HSC-4 cells, 25% in HSC-2 cells, and more than 40% in Ca9-22 [41]. Butyrate at 1.0 and 3.0 mM also reduced cell viability to around 50 and 90%, respectively [39]. In another study, 5 mM butyrate increased the number of dead cells and induced more lactate dehydrogenase release than LPS in primary gingival epithelial cells [37]. Likewise, acetate and propionate showed cytotoxic effects to HSC-2 cells after 24 h stimulation at 10 mM [36]. Epithelial cells were also exposed to different concentrations of “mixed SCFA” for up to 48 h in two studies [43,44]. At equimolar concentrations of 25 mM acetic, formic, lactic and propionic acids, cell viability was reduced by 95% at 48 h [43]. When the SCFA were considered separately, propionate was the most effective in reducing cell viability with 61% of inhibition, and acetate was the least effective with 10% of inhibition. Cells exposed to 12.5 mM mixed SCFA or less recovered shortly after acids removal [43]. Moreover, a ^3^H-thymidine incorporation assay revealed that, at 0.5 mM, the inhibition ranged from 3% for lactic acid to 53% for butyric acid after 30 h, and with 72% of inhibition for the 2 mM of mixed SCFA [44]. Interestingly, two studies showed a slight increase in the viability of Ca9-22 cells at low concentrations of butyrate, 0.02 mM [39] and 0.3 mM [41]. Taken together, these observations suggest that SCFA negatively affect the viability of oral epithelial cells at mM concentrations found in the crevicular fluid of patients affected by periodontitis [8].

Cell morphology changes: Consistent with the decrease in cell viability, morphological changes on human oral epithelial cells after SCFA exposure were reported [37,39,40]. Primary oral epithelial cells became swollen and flat with a blurred cellular contour when treated with 5 mM butyrate for 48 h, being different from LPS-treated cells, which presented cuboidal shape and a big nucleocytoplasmic ratio [37]. SEM and TEM revealed that cells presented pores ranging from 200 nm to 800 nm, decreased number of microvilli on the cellular membrane, organelles change and large bubbles on the surface in butyrate-treated cells. Cells grown in the presence of 3.0 mM butyrate for 24 h revealed morphological changes such as cell rounding [39,40]. Together, these morphological changes are signs for necrosis and apoptosis further supporting the view that SCFA disturbed the morphological integrity of the oral epithelial cells in vitro.

Cell death mechanisms: In support of the proposed apoptotic mechanisms, fragmented DNA was released from human oral epithelial cells after SCFA exposition for up to 48 h [44]. Moreover, in Ca9-22 cells, 10 mM butyrate increased caspase-3 activity, phosphatidylserine redistribution and bcl-2 down-regulation [33]. Butyrate also changed markers representing autophagy including microtubule-associated protein 1 light chain 3 (LC3) and 3-methyladenine. This observation is supported by findings that AMPK-dependent LC3 induction plays a role in butyrate-induced cell death [42]. Butyrate was also proposed to trigger cell death by pyroptosis where the intracellular contents released after rupture of the membrane may provoke an inflammatory response [37]. Taken together, SCFA may decrease cell viability by activating a series of cellular event that include apoptosis, autophagy and pyroptosis.

Regulation of adhesion and other molecules: Considering that a decrease in viability is associated with a loss of cell integrity, the expression of cytoskeletal and adhesion proteins was evaluated. In support of this assumption, butyrate increased the epithelial permeability for FITC-dextran and decreased the expression of connexin 26 and 43, adherence junction gene CDH1, junctional adhesion molecule-1, claudin-1 and 4, as well as desmoglein-1 and desmocollin-2 [37]. Interestingly, butyrate and propionate increased the keratins in the human oral epithelial cells, mainly keratin K17, further verified with immunohistochemistry in gingiva explants [38]. Podoplanin, a transmembrane glycoprotein related to the progression of human squamous cell carcinoma, was also increased in the presence of butyrate for HSC-2 and HSC-3 cell lines. Markers for epithelial–mesenchymal transition, E-cadherin and vimentin, however, showed inconsistent results [41]. Moreover, two studies reported an increase in intercellular adhesion molecule-1 (ICAM-1) expression with 3 mM butyrate in Ca9-22 cells, while integrins α6 and β4 were decreased [39,40]. By contrast, butyrate greatly suppressed ICAM-1 in HSC-2 cells under basal and inflammatory conditions [36]. It should be mentioned that butyrate but not LPS critically increased CXCL8, CCL2, CXCL10, and TNFα in primary human gingival epithelial cells. Butyrate also decreased CCL5 and other genes in this setting [37]. These data suggest that butyrate not only affects cell viability as the strong regulation of intercellular junctional and other proteins might be an independent effect of the SCFA on oral epithelial cells.

### 2.5. Certainty in Cumulative Eevidence and Risk of Bias across Studies

The certainty in cumulative evidence assessed by GRADE was considered low for the SCFA effects on cell death of human oral epithelial cells [33,37,39,40,41,42,43,44] and modulation of intercellular junction proteins expression [37,38,39]. For SCFA effects on the transmigration of immune cells [36,39,40], increasing on podoplanin expression in human oral epithelial cells [41], and reduction on butyric acid detrimental effects by sodium bicarbonate [40], the certainty in cumulative evidence was judged very low. According to the GRADE tool, the risk of bias across studies was considered as “serious” due to the presence of bias on the methodology in most of the studies screened. Summary of findings and further explanations with regard to evidence appraisal are presented in Table 3.

## 3. Discussion

The aim of this systematic review was to gather the in vitro evidence on the effects of SCFA in human oral epithelial cells in the context of periodontal disease. The results of this study revealed that SCFA, especially butyrate, might be detrimental to the epithelial barrier integrity. This may be explained by the contribution of SCFA to epithelial detachment, cell death, as well as to an altered junctional proteins expression. Although an influence of butyrate on ICAM-1 expression was reported [36,39,40], the overall effects of SCFA on leucocyte transmigration remain to be elucidated. Moreover, the evidence regarding the association of SCFA with external conditions such as nutrient-deficient environment and inflammation, the application of sodium bicarbonate or the relationship with oral carcinoma progression, is still insufficient to draw strong conclusions [36,40,41,42].

The present systematic review included only oral epithelial cells with human origin, thus excluding studies using animal-derived cell lines. Likewise, in vitro investigations with epithelial cells not derived from the oral cavity were excluded. Nevertheless, those findings may provide some additional clues. While growth inhibition by SCFA in HeLa and Vero epithelial cells [46,47] as well as human skin epithelial cells and porcine epithelial rests of Malassez cells were reported [48], cytotoxicity of SCFA was not observed in urothelial-derived cells at concentrations that were even higher than in the periodontal crevicular fluid [49]. Even though these studies may suggest some effects in a periodontal scenario, the use of different cell types limit the interpretation of the data.

Thus far, only few studies have been able to confirm the in vitro findings in preclinical and clinical research [37]. Histological images of gingival tissue after butyrate treatment revealed a reduced junctional epithelium layer, with cells presenting large intercellular space. Differences in gingival epithelial cells shape and a weaker staining with E-cadherin compared to control or LPS groups were also detected. Even though clinical studies have not yet been able to confirm detrimental effects of SCFA on the periodontal epithelium, non-surgical treatment of patients with periodontitis reduced the concentrations of SCFA in the crevicular fluid [8]. Thus, the current knowledge of SCFA effects on oral epithelium mainly remain at the level of in vitro research.

The possible role of other cell types as target for SCFA and their impact on periodontal disease requires further research. The first indications of butyrate and propionate as toxic metabolites for the periodontium came from experiments in human and mouse gingival fibroblasts [31]. Although healthy gingival fibroblasts were minimally affected by butyrate-induced apoptosis [50], inflamed gingival fibroblasts from adults with periodontitis were susceptible to mitochondria- and caspase-dependent apoptosis in butyrate-treated protocols [51]. Another study demonstrated that long-term exposure of human gingival fibroblasts to butyrate induced cytostasis and apoptosis, via activation of caspase-9 and caspase-8 (intrinsic and extrinsic pathways, respectively), accompanied by an upregulation of inflammatory cytokines [32]. Based on in vitro and preclinical data, SCFA may promote neutrophil migration through increased L-selectin expression and cytokine-induced neutrophil chemoattractant-2alphabeta release [52]. Short-chain fatty acids, however, did not alter leukocyte migration in an infectious site protocol but exerted detrimental effects by decreasing the phagocytic capacity of neutrophils along with modulating the production of inflammatory mediators [53]. Regarding lymphocytes, a study supports the role of SCFA, particularly butyrate, in the pathogenesis of periodontal diseases by suppressing T-cells proliferation and increasing IL-1β production by monocytes [54]. Butyrate also induced apoptosis in human T-lymphocytes by activating caspases signaling [55]. Overall, SCFA produced by periodontal pathogens seem to provoke harmful effects on immune and periodontal-native cells impairing the host response against bacterial invasion.

Some of the studies included in this systematic review investigated the underlying mechanisms by which SCFA may operate on epithelial cells. Two main possibilities were raised: a receptor-mediated pathway by sensing GPR41 and GPR43 (free fatty acid receptors; FFA3 and FFAR2, respectively) as well as other receptors from this family, or via inhibition of histone deacetylase (HDAC) by altering DNA transcription. A study compared both mechanisms by exposing HSC-2 cells to a GPR43 agonist and trichostatin A, a HDAC inhibitor. GPR43 produced a robust inhibition of the cytokine-induced expression of ICAM-1, whereas trichostatin A failed to reduce the cytokine-induced expression of ICAM-1 [36]. Conversely, the HDAC inhibition pathway was assessed in another study and the acetylation of histone H3 increased in butyrate treated cells [42]. Findings with other cell types also found contradictory results. Acetate promoted intestinal IgA responses mediated by GPR43 sensing [56] while SCFA inhibited HDAC activity in neutrophils [53]. What remains to be answered is: what is the signaling mechanism responsible for the decrease in epithelial cells viability? In vitro evidence suggests, for example, that propionate and cisplatin synergistically induce apoptosis of HepG2 hepatocellular carcinoma cells by increasing expression of TNF-α via reduction of histone deacetylases through GPR41 signaling [57]. GPR41 activation was also considered a compensatory mechanism to counter the increase in histone H3 acetylation by butyrate [58]. Others reported that butyrate and propionate increase apoptosis of neutrophils by factors other than GPRs [59]. Altogether, the operative mechanisms, and the particular involvement of GPRs and HDACs to exert the detrimental in vitro effects of SCFA on periodontal cells remain to be elucidated.

A complex issue that is worthy of mention is the relation between the concentrations of SCFA applied to cells in the experiments and the observed effects. Even when comparing the same molecule, the dose used in most of the included investigations largely varied, for instance, from 3 mM [39,40] to 20 mM [41] of butyrate for expression analyses. The treatment regime was also an aspect of considerable heterogeneity, even for similar assays across the included articles. Still, the multiple sources of SCFA among studies may have an impact on their outcomes. Guidelines and registration of protocols even for in vitro investigations may mitigate these discrepancies. Furthermore, using concentrations compatible to those found in patients with periodontal disease, approximately 10mM for propionate and 3mM for butyrate [49,60], would facilitate the translation and interpretation of the results, and therefore should be encouraged in further investigations. Despite these disparities, a dose–response effect could be observed in most of the included studies, indicating some consistency between the results observed.

A number of limitations of this review should be acknowledged. Although in vitro studies indicate that SCFA may have a negative effect on periodontal epithelial cells, in vitro models reproduce only a restricted aspect of the complex interaction between host-response and bacterial aggression that may culminate in periodontal disease. Clinical studies support the hypothesis that gut-derived SCFA arriving in the oral cavity via the blood stream may produce a decrease on periodontal parameters and inflammation, which indicate a probable protective effect in those cases [18,19,61]. Moreover, the initiation and progression of periodontal diseases involve a large number of cell types, such as fibroblasts, leukocytes, and lymphocytes, among others, and the present systematic review only covered cells from the oral epithelium. Therefore, the results of this systematic review are limited to an in vitro scenario and thus should be interpreted cautiously. Preclinical models provide a differential view on the role of SCFA in health and disease. For example, butyrate protected against oxidative stress in human nucleus pulposus cells by elevating peroxisome proliferator-activated receptor gamma-regulated klotho expression [62], and reduced infarct volume and improved neurological function of middle cerebral artery occlusion via GPR41 signaling [63]. Butyrate also exerts protective effects on colitis [64,65] and bone loss [66]. Thus, care should be taken regarding the overall impact of SCFA on the integrity of the periodontal epithelium based on in vitro studies.

Future research should investigate the mechanisms of epithelial cell death associated to SCFA, as well as the influence of those metabolites to intercellular junction proteins expression. For example, knockout strains for GPR41, GPR43, GPR109A, and GPR43/109A are available to identify which receptor mediates the effects of the SCFA as reported in cardiovascular research [67]. Similar strategies could be applied to identify the impact of GPRs on ligature-induced plaque accumulation and inflammatory osteolysis in mice [68]. Moreover, investigation of the influence of SCFA on the transmigration of leucocytes and immune cells is strongly recommended due to its importance on the development and progression of periodontal disease by considering for instance the use of GPR knockout models. ATG5 and ATG7 knockout mouse models would also help to understand the possible role of SCFA on autophagy [69], as for example, butyrate induced mTOR pathway-activated autophagy in bladder cancer cells [70] and nasopharyngeal carcinoma cells [71]. Even more exciting, with respect to the present review, would be to recommend research with conditional knockout models for epithelial cells [69]. Thus, the in vitro effects on cell viability should not be restricted to apoptosis as SCFA also affect autophagy. What has not been investigated so far is the impact of SCFA on oral cell necroptosis, a mechanism that has been linked to periodontal homeostasis [72]. It seems likely that SCFA trigger necroptosis in oral squamous carcinoma cells as for example chidamide, a HDAC inhibitor which induces necroptosis in Jurkat and HUT-78 cells [73]. Thus, future in vitro research needs to identify the signaling cascade on how the SCFA exert their detrimental in vitro effects in oral epithelial cells and whether the in vitro findings can be translated in preclinical models with the overall aim to identify SCFA as molecular targets in periodontal research and related fields.

Since a meta-analysis was not possible due to the heterogeneity of the included studies, a standardization of fatty acids used, molar concentration, protocol of intervention, and assays applied is strongly recommended. The inclusion of primary gingival epithelial cells is also mandatory in future investigations on the topic. Moreover, prospective in vivo research could give a new perspective to the understanding of how SCFA disturb periodontal tissues with the presence of the confounding factors of an inflammatory process. Overall, and despite these limitations, the present systematic review provides a comprehensive summary of the current in vitro evidence on the effects of SCFA on human oral epithelial cells and may inspire further research to better understand the influence of these metabolites on periodontal health and the triggering of the disease process.

## 4. Materials and Methods

### 4.1. Study Design

This systematic review was conceived at the Department of Oral Biology of the Medical University of Vienna, Austria. The study protocol was developed and discussed by the authors based on the Cochrane Handbook for Systematic Reviews of Interventions [74] and Preferred Reporting Items for Systematic Review and Meta-Analysis (PRISMA-P) checklist [75] guidelines. We registered this systematic review on Open Science Framework registration platform (doi:10.17605/OSF.IO/THXQM) on 12 May 2020.

### 4.2. Focused Question

Based on the PICOS acronym [76], the following focused question was defined: In in vitro studies (S), what are the effects of short-chain fatty acids (I), as compared with untreated groups (C), in human oral epithelial cells (P) considering the potential impact on molecular and cellular parameters related to periodontal disease (O)? Details are presented in Table 4.

### 4.3. Eligibility Criteria

As the aim of this systematic review was to evaluate the in vitro effects of SCFA on human oral epithelial cells that may influence periodontal disease, original in vitro studies that could demonstrate this association were allowed. No restriction of time, status of publication or language were applied. The exclusion criteria were:Studies in which other cells apart from human oral epithelial cells were used, such as cells of animal origin, from other organs (lung, skin, liver, kidney, colon, etc.), and derived from other tissues (fibroblasts, neutrophils, macrophages, etc.);Studies in which SCFA were not used as a treatment to epithelial cells or studies without a control group;Studies in which the molecular and cellular effects of SCFA on cells potentially related to periodontal disease were not evaluated;Clinical and animal studies;Pilot studies and research projects.

### 4.4. Search Strategy

The electronic database search strategy was developed by the authors based on the PICOS structure using MeSH terms and key words. Search strategy was developed for PubMed and adapted for each of the electronic databases used: PubMed, Scopus, Web of Science, Literature of Latin American and Caribbean Health Sciences (LILACS) and Embase. Additionally, unpublished literature was verified in Open Grey, ProQuest, and Google Scholar databases (Appendix A). The electronic search was performed up to 10 June 2020. In addition, a manual search using the references of the identified records was performed. Experts were also consulted by email to indicate additional references. All records founded were exported to a reference manager software (Mendeley Desktop, Elsevier, London, UK) and duplicates were removed.

### 4.5. Selection of Studies

Studies were selected in two phases. In phase 1, titles and abstracts were independently selected by two reviewers (G.L.M. and F.J.S.), and all articles that did not meet the eligibility criteria were removed. In phase 2, the same reviewers applied the eligibility criteria to the full text of studies. Any disagreement between the two reviewers was resolved by discussion and consensus, moderated by a third party (C.A.M.B.). Articles that met the eligibility criteria were then proceeded for data extraction.

### 4.6. Data Extraction

Data were collected independently and in duplicate by two investigators (G.L.M. and F.J.S.). A third reviewer (C.A.M.B.) was consulted in case of disagreements. The following data were extracted for each included study: author, year of publication, country, cell types, origin of cell lines, interventions and controls used, concentration of treatment, regimes of interventions, methods applied, results, main conclusions, and possible clinical implications on periodontal disease. If necessary, corresponding authors of the included studies were contacted via email for clarification of any missing information and/or clarification of methodology and results.

### 4.7. Assessment of Risk of Bias in Individual Studies

Two reviewers (G.L.M. and F.J.S.) independently performed the risk of bias assessment adapting the tool suggested by the World Cancer Research Fund/ University of Bristol for cell line studies [45]. Questions could be answered as “yes”, “no”, “unclear” or “not applicable”. Included studies were categorized as “low”, “moderate” or “high” risk of bias, according to the following: (a) low risk of bias, if studies reached more than 70% scores of “yes”; (b) moderate risk of bias, if “yes” scores were between 50% and 69%; and (c) high risk of bias, if “yes” scores were below 49%. Disagreements between the authors were discussed with a third reviewer (C.A.M.B.) until consensus was achieved.

### 4.8. Summary Measures and Synthesis of Results

Information concerning cell viability after SCFA treatment, regulation of protein expression level in human oral epithelial cells and other measures were considered during evaluation of outcomes. A meta-analysis was planned, if the data from the included studies were considered relatively homogeneous.

### 4.9. Grading of Cumulative Evidence and Risk of Bias Across Studies

Two authors (G.L.M. and F.J.S.) methodically appraised the selected studies according to the grading of recommendations assessment, development, and evaluation (GRADE) method to judge the certainty in cumulative evidence [77]. They evaluated risk of bias, inconsistency, indirectness, imprecision, and publication bias, and categorized the outcomes across the included articles as “high”, “moderate”, “low” or “very low” quality of evidence, according to their analysis of each study. For the risk of bias across studies a decision between “not serious”, “serious” and “very serious” was given based on GRADE considering all the outcomes evaluated. When the two authors responsible for evaluating the studies did not reach a consensus, a third author (C.A.M.B.) intervened to make a final decision. Studies were classified with comparable baselines, which were compared as a randomized controlled study, according to the work of Pavan et al. published in 2015 [78].

## 5. Conclusions

Overall, the findings of this systematic review suggest the following conclusions:SCFA impair the viability of oral epithelial cells at mM concentrations via apoptosis, autophagy, and pyroptosis.SCFA impair the integrity and presumably the transmigration of leucocytes through the epithelial layer by changing junctional and adhesion protein expression, respectively.SCFA affect the expression of chemokines and cytokines in oral epithelial cells.

## Figures and Tables

**Figure 1 ijms-21-04895-f001:**
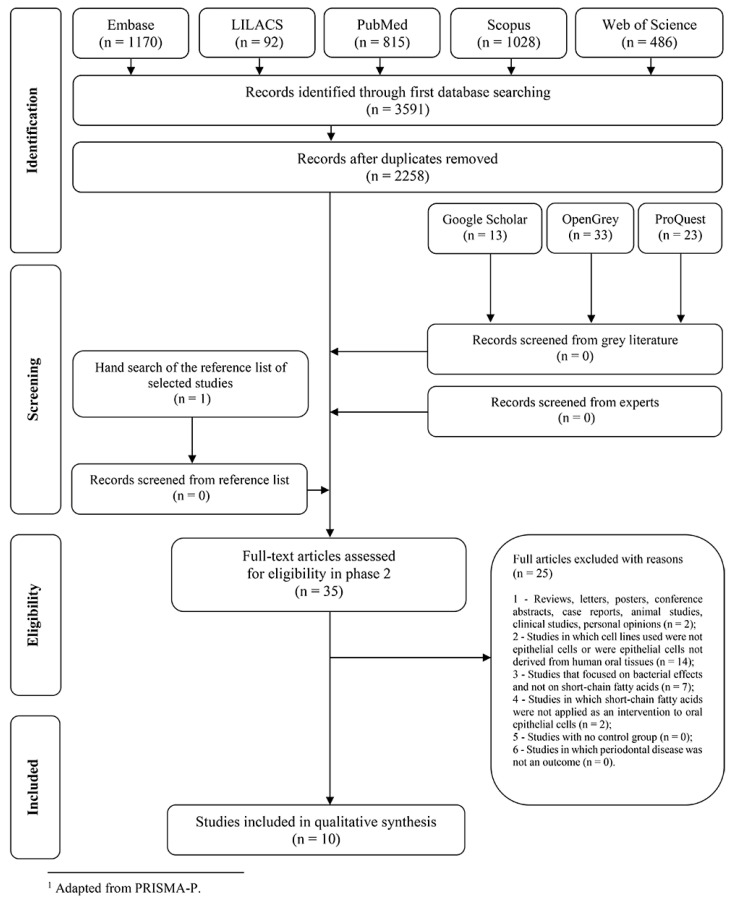
Flow Diagram of literature search and selection criteria.

**Figure 2 ijms-21-04895-f002:**
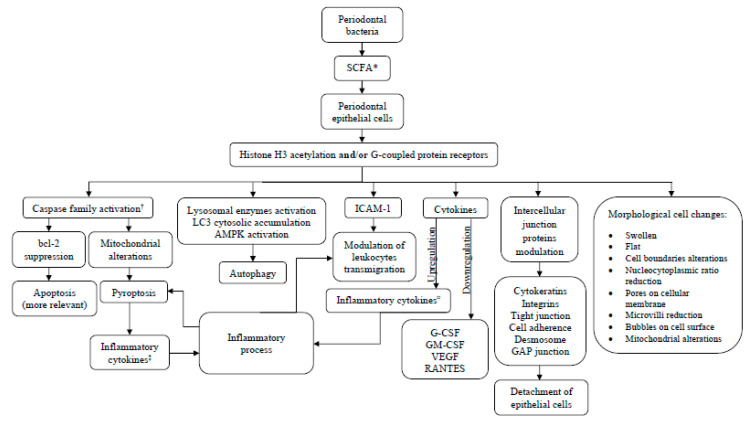
Schematic flowchart of potential molecular mechanisms of SCFA on oral epithelial cells.

**Table 1 ijms-21-04895-t001:** Summary of descriptive characteristics of included articles (*n* = 10).

Author (Year); Country	Cell Type (Origin)	Treatment/ Concentration	Treatment Regime	Methods: Assays	Results	Main Conclusions	Clinical Application
Evans et al. (2017); Japan [42]	Ca9-22 (HOSCC)	Sodium butyrate/ 5 mM; RPMI1640 (control)	48 h incubation ^a^; 24 h incubation ^b^; 0, 2, 4 and 6 h incubation ^c^; 0, 2, 4 and 8 h incubation ^d^	SYTOX Green cell death assay ^a^; Caspase-3 assay ^b^; Caspase-3/7 assay ^b^; qRT-PCR analysis ^c^; Western blot ^b,d^	Butyrate induced cell death in starved and sufficient nutrient conditions.Although butyrate treatment induced about 10 times higher caspase-3 activity (as a measure for apoptosis) compared to the non-treated cells, butyrate-induced apoptosis accounted for only 29.4 or 12.3% of total cell death induced by butyrate in the starved and fed conditions, respectively.The combined effects of histone H3 acetilatilation, AMPK activation, and LC3 upregulation during starvation and butyrate exposure resulted in induction of cell autophagy.	Starvation enhances butyrate induced cell death and autophagy of gingival epithelial cells.	The combined stimuli of butyrate exposure and cell starvation is may be involved in tissue destruction at the dentogingival junction.
Liu et al. (2019); China [37]	Primary gingival epithelial cells (freshly isolated)	Sodium butyrate/ 10 mM; LPS/ 1 μg/mL; KGM (control)	48 h incubation	Transepithelial electrical resistance; FITC-dextran transport assay; Flow cytometry; qRT-PCR analysis; Western blot; Immunohisto-chemestry; Immuno-cytochemistry; Im-munofluorescence; FE-SEM; TEM	Butyrate had a stronger effect on cell membrane damage than LPS, altered cell morphology, increased cell death and down regulated the intercellular junction markers. Sodium butyrate, rather than LPS, increased inflammatory chemokines of periodontitis, such as IL8 and MCP1, suggesting a role as virulence factor in gingival epithelial cells. During butyrate treatment, mitochondria lost their morphologies, which may indicate a triggering for the cascade of cell pyroptosis.	Butyrate rather than LPS subverts the gingival epithelial barrier function by triggering gingival epithelial cell pyroptosis and downregulating the expression of intercellular junction proteins.	Butyrate acts in the destruction of the gingival epithelial barrier, and may play a role in initiating periodontitis.
Magrin et al. (2020); Austria [36]	HSC-2 (HOSCC); TR146 (HOSCC); Primary gingival epithelial cells (freshly isolated)	Acetate/ 10 mM; Propionate/ 10 mM; Butyrate/ 10 mM; DMEM (control)	24 h incubation, 3 h exposure to TNFα + IL1β ^a^; 24 h incubation (butyrate only), 3 h exposure to TNFα + IL1β ^b^; 24 h incubation (butyrate only), 30 min exposure to TNFα + IL1β ^c^; 1 h exposure to TNFα + IL1β, 3 h incubation (butyrate only) ^d^	MTT assay; qRT-PCR analysis ^a,d^; Western blot ^b^; Immuno-fluorescence ^c^	Butyrate suppressed in a dose-dependent manner the cytokine-induced ICAM1 expression in HSC-2 and primary gingival epithelial cells but not in TR146 cells. Acetate and propionate failed to cause a significant suppression of cytokine-induced ICAM1 expression. Butyrate inhibited the nuclear translocation of p65 on HSC-2 cells. Butyrate failed to reverse cytokine-induced ICAM1 increase in HSC-2 cells for an acute inflammation protocol.	Butyrate but not acetate or propionate attenuates the cytokine-induced ICAM1 expression in oral epithelial cells.	Butyrate can modulate epithelial cell responses in the inflamed periodontium and thereby possibly influencing the ICAM1-dependent transmigration of leucocytes and immune cells.
Miyazaki et al. (2010); Japan [41]	Ca9-22 (HOSCC); HSC-2 (HOSCC); HSC-3 (HOSCC); HSC-2 (HOSCC)	Sodium butyrate/ 0.3, 2.5 and 20 mM; RPMI1640 (control)	24 h incubation; 8 h incubation ^a^; 4, 8, 12 and 24 h incubation ^b^	MTT assay; “Scratch” assay ^a^; qRT-PCR analysis ^b^; Western blot	The proliferative activities of HSC-2,-3 and -4 cells decreased with butyrate in a dose-dependent manner, whereas in Ca9-22 cells slightly increased in 0.3 mM concentration. Expression of podoplanin (oral cancer biomarker) was enhanced by butyrate in HSC-2 and HSC-3 cells. Cell migration was inhibited in the presence of butyrate for HSC-2 and HSC-4 cells.	Sodium butyrate increases podoplanin expression and cell migration in certain HOSCC cell lines, suggesting that the progression of periodontal disease may promote the progression of oral squamous cell carcinomas via a podoplanin-dependent pathway.	It is suggested an association of butyrate produced by periodontopathic bacteria with the progression of oral cancers.
Pöllänen & Salonen (2000); Finland [38]	Immortalized human oral epithelial cells (gingival keratinocytes); Primary gingival epithelial cells (freshly isolated)	Sodium butyrate/ 8 mM; Sodium propionate/ 8 mM; KBM (control)	24 h incubation	Western blot; Immunohisto-chemestry	Propionate did not affected keratinocyte cell numbers, whereas butyrate reduced cell numbers by about 30%. The expression of the cytoplasmic keratin K17 was markedly increased with propionate and especially butyrate, further confirmed in primary gingival epithelial cells.	Butyrate and propionate increase the relative amount of keratin proteins in the cells, most strikingly keratin K17, a protein related to periodontal pocket formation.	The increased expression of K17 after SCFA exposure may contribute to detachment of the junctional epithelium from tooth surface and to the formation of periodontal pockets.
Sorkin & Niederman (1998); USA [44]	Immortalized human oral epithelial cells (HPV-transformed cells)	“Mixed SCFA” (butyric, propionic, acetic and lactic acids)/ up to 20 mM in a mixture of equimolar concentrations of each acid; KGM or KGM + CaMg medium (control)	30 h incubation ^a^; 9 days incubation (medium changed every other day) ^b^; 2, 8 and 48 h incubation ^c^	Cell proliferation assay ^a^; Cell survival assay ^b^; ELISA assay ^c^	SCFA decreased cell proliferation and cell survival in a dose-dependent manner. At all time points, ELISA assay showed more DNA in the cytoplasmic extracts and in the supernatants, suggesting that many of the cells may be initiating apoptotic cell death.	SCFA decrease gingival epithelial cell proliferation and increase apoptosis and necrosis. These effects were dose- and acid-dependent.	By decreasing the proliferative capacity of the gingival epithelium, SCFA could increase epithelial permeability over time, increasing crevicular fluid flow, and bacterial penetration.
Takigawa et al. (2008a); Japan [39]	Ca9-22 (HOSCC)	Butyric acid/ 3 mM; MEM (control)	24 h incubation; 0 to 7 days incubation ^a^; 6 h incubation ^b^	SEM; Flow cytometry; Cell proliferation assay ^a^; qRT-PCR analysis ^b^	Cell growth was inhibited in a dose-dependent manner after butyric acid treatment. Butyric acid exposure for 6 h increased ICAM1 expression, decreased integrin α6 and β4 levels, and reduced cell viability. Moreover, morphological changes were observed after 24 h.	Butyric acid inhibits cell growth, reduces cell viability, suppress integrin levels and alters the expression of ICAM1 in gingival epithelial cells.	Butyric acid in periodontal pockets may augment inflammatory cell migration and disable the tight attachment among epithelial cells, leading to bacterial invasion and periodontal damage.
Takigawa et al. (2008b); Japan [40]	Ca9-22 (HOSCC)	Butyric acid/ 3 mM; Butyric acid + NaHCO_3_/ 3mM; MEM (control)	24 h incubation; 6 and 24 h incubation ^a^	SEM; Flow cytometry; Trypan blue exclusion assay (cell viability) ^a^; qRT-PCR analysis ^a^	Cell viability after butyric acid exposure was lower than that of butyric acid plus NaHCO_3._ ICAM1 expression was increased with butyric acid alone and suppressed in the presence of NaHCO3 after 6 h incubation. However, no differences were detected after 24 h.	NaHCO_3_ improves cell viability and inhibit ICAM1 expression increasing in butyric acid treated cells.	NaHCO_3_ may have a useful therapeutic application to reduce butyric acid damage on periodontal tissue.
Tsuda et al. (2010); Japan [33]	Ca9-22 (HOSCC)	Sodium butyrate/ 10 mM; RPMI1640 (control)	48 h incubation; 24 h incubation ^a^; 0, 4, 8, 12 and 24 h incubation ^b^; 2, 4 and 8 h incubation ^c^; 8 h incubation ^d^	Microscopic observation; SYTOX Green cell death assay; Annexin V–FITC assay; Caspase-3 assay ^a^; qRT-PCR analysis ^b^; Western blot ^c^; Fluorescence microscopy ^d^	Butyrate stimulation induced apoptotic cell death in a dose- and time-dependent manner. Treatment with 3-methyl-adenine and CA-074 Me (proteins of cell autophagy), further confirmed with LC3 tests, suppressed butyrate-induced cell death, sugges-ting an autophagic pathway.	Butyrate induces the death of gingival epithelial cells primarily via caspase-independent autophagy and partly via apoptosis.	Butyrate may play an important role in killing gingival epithelial cells and breakdown the integrity of the front-line epithelial barrier of gingival tissues.
Zhang & Kashket (1997); USA [43]	Immortalized human oral epithelial cells (HPV-transformed cells)	“Mixed SCFA” (acetic, formic, lactic and propionic acids)/ up to 100 mM in a mixture of equimolar concentrations of each acid; NaCl (salt control); KGM (control)	2, 16, 24, 40, 48 and up to 64 h incubation; 16 h incubation, change medium to KGM for another 16, 24, 40 or 48 h incubation (reversibility protocol)	MTT assay; Trypan blue exclusion assay	Cell growth was completely inhibited in the presence of 50 mM or above of mixed SCFA. Few cells were dead in the control or in cultures with up to 20 mM mixed SCFA. Cells exposed to 12.5 mM SCFA or less began to grow shortly after acids removal. Cells exposed to 25 mM mixed SCFA began to grow only after a recovery period of about 16 h.	Cell growth is progressively inhibited with increasing concentrations of SCFA.	SCFA can damage the integrity of gingival epithelium in situ.

RPMI1640 = Roswell Park Memorial Institute 1640 medium; qRT-PCR = reverse transcription polymerase chain reaction; KBM = Keratinocytes basal medium; AMPK = 5′ adenosine monophosphate-activated protein kinase; LC3 = microtubule-associated protein 1 light chain 3; LPS = Lipopolysaccharide; MCP1 = monocyte chemoattractant protein 1; KGM = Keratinocytes growth medium; FE-SEM = Field emission scanning electron microscopy; TEM = Transmission electronic microscopy; HOSCC = Human oral squamous cell carcinoma; DMEM = Dulbecco’s modified Eagle medium; HPV = Human papilloma virus; MEM = Minimum essential medium; SEM = Scanning electron microscopy; SCFA = Short-chain fatty acids. Superscript letters in the same row are associated (treatment regime and assay applied).

**Table 2 ijms-21-04895-t002:** Risk of bias in individual studies assessed by the adapted tool of the World Cancer Research Fund/ University of Bristol for cell line studies (*n* = 10).

Study	Q1	Q2	Q3	Q4	Q5	Q6	Total (% Score Yes)	Risk of Bias
Evans et al. (2017) [42]	Yes	Unclear	No	Yes	Yes	Not applicable	50,0	Moderate
Liu et al. (2019) [37]	Yes	Yes	Yes	Yes	Yes	Not applicable	83,3	Low
Magrin et al. (2020) [36]	Unclear	Yes	Yes	Yes	Unclear	No	50,0	Moderate
Miyazaki et al. (2010) [41]	Yes	Unclear	No	Yes	Yes	Yes	66,6	Moderate
Pöllänen & Salonen (2000) [38]	Yes	Unclear	Yes	Yes	Unclear	No	50,0	Moderate
Sorkin & Niederman (1998) [44]	Unclear	Yes	No	No	Yes	Not applicable	33,3	High
Takigawa et al. (2008a) [39]	Yes	Yes	No	Yes	Yes	Not applicable	66,6	Moderate
Takigawa et al. (2008b) [40]	Yes	No	No	Yes	Yes	Not applicable	50,0	Moderate
Tsuda et al. (2010) [33]	Yes	Yes	No	Yes	Unclear	Not applicable	50,0	Moderate
Zhang & Kashket (1997) [43]	Unclear	Yes	No	No	Yes	Not applicable	33,3	High

Questions used for assessing the risk of bias of the included studies were: (Q1) Have the cells been obtained from a validated repository that guarantees cell verification or have the cells been appropriately independently verified? (Q2) Have sufficient biological and technical repeats of the experiments been conducted and were appropriate controls included? (Q3) Were primary cell lines from periodontal epithelium used in the study? (Q4) Were concentration of the molecules in the intervention group and treatment regime comparable to other studies? (Q5) Was selective reporting avoided?—question if only selected results from different experiments or cell lines were reported. (Q6) Were cell lines from different types compared?

**Table 3 ijms-21-04895-t003:** Summary of findings based on GRADE certainty of evidence assessment.

Outcomes	Impact	Number of Studies	Certainty of the Evidence (GRADE)
SCFA induce cell death of human oral epithelial cells	Increase in periodontal epithelial barrier destruction	8 studies	⊕⊕◯◯ LOW ^a^^,^^b^^,^^c^
SCFA modulate intercellular junction proteins expression	Contribute to epithelial detachment and periodontal pocket formation	3 studies	⊕⊕◯◯ LOW ^a^^,^^b^^,^^e^
SCFA influence the transmigration of leucocytes and immune cells	Modulation on immune cell migration	3 studies	⊕◯◯◯ VERY LOW ^a^^,^^b^^,^^c^^,^^d^^,^^e^
Butyrate increases podoplanin expression in human oral epithelial cells	Butyrate produced by periodontopathic bacteria is associated with the progression of oral cancers	1 study	⊕◯◯◯ VERY LOW ^b^^,^^e^^,^^f^^,^^g^
Sodium bicarbonate reduces butyric acid detrimental effects	Sodium bicarbonate have a useful therapeutic application to reduce butyric acid damage on periodontal tissue	1 study	⊕◯◯◯ VERY LOW ^b^^,^^c^^,^^e^^,^^f^^,^^g^

GRADE Working Group grades of evidence: High certainty = we are very confident that the true effect lies close to that of the estimate of the effect; Moderate certainty = we are moderately confident in the effect estimate: The true effect is likely to be close to the estimate of the effect, but there is a possibility that it is substantially different; Low certainty = our confidence in the effect estimate is limited: The true effect may be substantially different from the estimate of the effect; Very low certainty = we have very little confidence in the effect estimate: The true effect is likely to be substantially different from the estimate of effect. Explanations: (a) Included studies presented high methodological and statistical heterogeneity; (b) There is no standardization of methods, doses and treatment regimens; (c) Studies from the same authors or conducted in the same institution presented similar results; (d) Collectively, studies presented a high risk of bias; (e) Estimates were not sufficiently supported by the presented experiments; (f) The study presented a moderate risk of bias; (g) Experiments with primary oral epithelial cells were not conducted.

**Table 4 ijms-21-04895-t004:** PICOS strategy.

PICOS	
**P**articipants	Human oral epithelial cells
**I**ntervention	Short-chain fatty acids
**C**omparison	No treatment with short-chain fatty acids (control)
**O**utcomes	Molecular and cellular parameters related to periodontal disease *
Types of **S**tudies included	In vitro studies

Based on PICO strategy for studies of intervention (Needleman IG, 2002; Higgins JPT, Green S. Cochrane Handbook for Systematic Reviews of Interventions Version 5.1.0. England: John Wiley & Sons, Ltd.; 2011). * Molecular and cellular parameters related to periodontal disease could be inflammatory cytokines expression, immune cells migration, cell death index, and others.

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
