# Peer review of "Effects of Short-Chain Fatty Acids on Human Oral Epithelial Cells and the Potential Impact on Periodontal Disease: A Systematic Review of In Vitro Studies"

_ijms, 2020, doi:10.3390/ijms21144895_

Round 1

Reviewer 1 Report

In the study titled “Short-chain fatty acids on human oral epithelial cells and the potential impact on periodontal disease: A systematic review of in vitro studies”, the authors performed a review of the available literature to evaluate the effect of SCFAs on oral epithelial cells in order to elucidate their impact on periodontal disease. This is a well-planned study.

(1) The authors should change the title to “Effect of short-chain fatty acids on human oral epithelial cells……..”

(2) There are many oddly phrased sentences of grammatical concern. The authors should proof-read to improve the English text.

(3)SCFA is written as SFCA in many places. Please maintain consistency throughout.

Author Response

Reviewer comments to the Authors:

Reviewer 1:

In the study titled “Short-chain fatty acids on human oral epithelial cells and the potential impact on periodontal disease: A systematic review of in vitro studies”, the authors performed a review of the available literature to evaluate the effect of SCFAs on oral epithelial cells in order to elucidate their impact on periodontal disease. This is a well-planned study.

AUTHORS: We thank the reviewer for his/her encouraging comments.

(1) The authors should change the title to “Effect of short-chain fatty acids on human oral epithelial cells……..”

AUTHORS: We accept the suggestion and the title has been changed as requested.

(2) There are many oddly phrased sentences of grammatical concern. The authors should proof-read to improve the English text.

AUTHORS: We agree with the reviewer, therefore, the revised version of our manuscript has been modified accordingly and proof read by a native speaker.

(3) SCFA is written as SFCA in many places. Please maintain consistency throughout.

AUTHORS: We apologize for that mistake. It was our oversight. Now the acronyms have been corrected accordingly.

Reviewer 2 Report

This review on SCFA effects on oral epithelium is much needed but the authors can explain and review a bit better. The whole focus is on the effects of SCFA. mainly butyrate on oral epithelium. While the methodology is aptly followed for a systematic review, I feel that the authors need to expand a bit more on potential mechanisms on the controversial nature of SCFAs on oral epithelial cells. I think the authors can present their findings in a graphical form, with a nice Figure illustrating the proposed mechanisms they comprehend through the review of all these studies. The other complex issue that has not been well addressed is the dose Vs effect. May be a separate section on SCFA dose Vs observed effects would bring that out. 

Can the authors also comment on the source of various SCFAs utilised in many of the studies? will that have an impact on outcomes? 

Overall, a good review that can be enhanced and presented better.

Author Response

Reviewer 2:

This review on SCFA effects on oral epithelium is much needed but the authors can explain and review a bit better. The whole focus is on the effects of SCFA, mainly butyrate, on oral epithelium. While the methodology is aptly followed for a systematic review, I feel that the authors need to expand a bit more on potential mechanisms on the controversial nature of SCFAs on oral epithelial cells. I think the authors can present their findings in a graphical form, with a nice Figure illustrating the proposed mechanisms they comprehend through the review of all these studies.

AUTHORS: We thank the reviewer for his thorough comments. As suggested, a new scheme (Figure 2) has been included in the Results section to graphically illustrate the potential mechanisms by which SCFA affect oral epithelial cells.

The other complex issue that has not been well addressed is the dose Vs effect. May be a separate section on SCFA dose Vs observed effects would bring that out.

AUTHORS: The reviewer has a point. We recognize that the SCFA dose may modify the observed effect on epithelial cells. However, we feel that dose Vs effect aspect is more suitable to be addressed and further discussed the Discussion section. Therefore, a completely new paragraph in the Discussion section to address that point.

Can the authors also comment on the source of various SCFAs utilized in many of the studies? Will that have an impact on outcomes?

AUTHORS: We appreciate the questions of the reviewer. Indeed, the source of SCFA may influence the effects on cells. Therefore, this point has been addressed and discussed in the Discussion section of the revised manuscript.

Overall, a good review that can be enhanced and presented better.

AUTHORS: We thank the reviewer for the encouraging comment.

Reviewer 3 Report

Thank you very much for submitting your research to International Journal of Molecular Sciences. Reviewer would like to make several comments on your article.

1.When did the authors registered the protocol?

We registered this systematic review on Open Science Framework registration platform.

2.Do the authors think there are enough data for the review?

3.Please provide the opinion from the authors regarding the diversity in the design among the studies.

4.Please provide a more detailed comments on the types of the cells.

5.Please consider the confounding factors in the analysis.

6.Please consider the performing the prospective studies to evaluate the characteristics.

7.You may add more data to make it more sound.

Thank you very much.

Author Response

RESPONSE LETTER

Reviewer 3:

Thank you very much for submitting your research to International Journal of Molecular Sciences. Reviewer would like to make several comments on your article.

  1. When did the authors registered the protocol? “We registered this systematic review on Open Science Framework registration platform.”

AUTHORS: We thank the reviewer for this question. The protocol approval was obtained the 12th May 2020. This information is now included in our manuscript. Although recommended, the registration of systematic reviews is not mandatory. In our case, we intended to register our study since the first discussions on the conceptualization phase. However, the registration of systematic reviews of in vitro studies is not possible in platforms such as PROSPERO and the authors were not aware of other platforms to perform this registration. We came across with the Open Science Framework registration platform when the systematic review had already started, and we registered as soon as possible. To prevent any inconsistencies between the protocol registration and the database search, we updated our search up to 10th June 2020. This information can be checked in our Materials and Methods, and Results sections, as well as in Figure 1 (PRISMA-P flowchart).

  1. Do the authors think there are enough data for the review?

AUTHORS: We feel that the amount of data compiled were sufficient for a systematic review, although only 10 articles were included. Furthermore, we believe that a systematic review on the topic is important in order to synthesize the available evidence.

  1. Please provide the opinion from the authors regarding the diversity in the design among the studies.

AUTHORS: We appreciate the suggestion of the reviewer and this aspect has now been discussed in the revised version of the manuscript. It can be checked in our Discussion section. In brief, there is a lack of standardization on fatty acids used, molar concentration, protocols of intervention and assays applied. This heterogeneity precluded a thorough comparison between the studies and a meta-analysis.

  1. Please provide more detailed comments on the types of the cells.

AUTHORS: Details about the cell types are presented in the Results section and on Table 2. We have also included in our Discussion section some details about the studies that used epithelial cells from other origins such as fibroblasts, leukocytes, among others as target for SCFA.

  1. Please consider the confounding factors in the analysis.

AUTHORS: We thank the reviewer for raising this issue. Now we have added a new paragraph in the Discussion section in which confounding factors are further discussed. Some included issues are the different concentrations of SCFA applied to cells in the experiments, the variations on the treatment protocol used in each investigation, the multiple sources of SCFA among studies and the dose-response effect. Moreover, other confounding factors have also been mentioned in the revised Discussion section such as the involvement of a large number of cell types, such as fibroblasts, leukocytes and lymphocytes, in the pathogenesis of periodontal disease.

  1. Please consider the performing the prospective studies to evaluate the characteristics.

AUTHORS: We thank the reviewer for the suggestion. Prospective in vivo research has a fundamental importance on the understanding of how SCFA impairs periodontal tissues in the presence of the confounding factors. This issue was therefore mentioned in our revised Discussion section.

  1. You may add more data to make it more sound.

AUTHORS: Thank you for your recommendation. The manuscript has been considerably changed according to the reviewers´ suggestions thereby improving the readability of the manuscript substantially. Moreover, we included a schematic flowchart (Figure 2) to illustrate the potential mechanisms by which SCFA may operate on oral epithelial cells in the context of periodontal disease. We hope the reviewer agrees with the changes.

Thank you very much.

AUTHORS: We appreciate your time and your worthy comments about our article.

Reviewer 4 Report

This is not a systematic review as the outcome measure(s) were not defined. "Periodontal parameters" is not an appropriate outcome measure. Periodontal parameters are CAL, PPD, BOP etc.

Title does not make sense as it sounds like SCFA are expressed on epithelial cells.

Grammar needs to be corrected.

The authors registered the protocol on 12/5/2020 and submitted the finished paper on 21/5/2020.

The topic lends itself to a narrative type of a review only.

Author Response

Reviewer comments to the Authors:

Reviewer 4:

This is not a systematic review as the outcome measure(s) were not defined. "Periodontal parameters" is not an appropriate outcome measure. Periodontal parameters are CAL, PPD, BOP etc.

AUTHORS: We agree with the reviewer that the term was misleading and apologize for presenting our outcome measures mistakenly. Therefore, we have changed it accordingly. Our intention was to cover cellular responses including proliferation, migration and differentiation, all of which may predict a possible clinical effect. In the revised version of the manuscript, we have explained the outcomes more precisely.

Title does not make sense as it sounds like SCFA are expressed on epithelial cells.

AUTHORS: We apologize if the title was misleading. We have changed the title to “Effects of short-chain fatty acids on human oral epithelial cells and the potential impact on periodontal disease: A systematic review of in vitro studies.”

Grammar needs to be corrected.

AUTHORS: The revised version of the manuscript has been substantially modified. In addition, the manuscript has been proof read by a native speaker in order to improve grammar and the readability of the manuscript.

The authors registered the protocol on 12/5/2020 and submitted the finished paper on 21/5/2020.

AUTHORS: The protocol approval was obtained the 12th May 2020. This information is now included in our manuscript. Although recommended, the registration of systematic reviews is not mandatory. In our case, we intended to register our study since the first discussions on the conceptualization phase. However, the registration of systematic reviews of in vitro studies is not possible in platforms such as PROSPERO and the authors were not aware of other platforms to perform this registration. We came across with the Open Science Framework registration platform when the systematic review had already started, and we registered as soon as possible. To prevent any inconsistencies between the protocol registration and the database search, we updated our search up to 10th June 2020. This information can be checked in our Materials and Methods, and Results sections, as well as in Figure 1 (PRISMA-P flowchart).

The topic lends itself to a narrative type of a review only.

AUTHORS: The reviewer has a point. However, we believe that there is a demand for a systematic approach to synthesize the existing knowledge of SCFA on cellular responses along with their plausible role in periodontal disease. It is quite likely that a narrative literature review would have omitted some studies since the search is neither systematic nor reproducible. Furthermore, the use of tools to evaluate the risks of bias and the certainty in cumulative evidence can certainly suggest some directions for future studies and research. Although only 10 articles met the eligibility criteria of our study, we retrieved several studies on the topic, enough for a systematic review.

Round 2

Reviewer 2 Report

All the corrections are relevant. The new Figure looks excellent.

Author Response

All the corrections are relevant. The new Figure looks excellent.

AUTHORS: Thank you very much for the encouraging comment. We appreciate your time and your valuable considerations.

Reviewer 3 Report

Thank you very much for submitting your revised research to International Journal of Molecular Sciences. Reviewer would like to make several comments on your article.

1.Please list the types of short-chain fatty acids and the short-chain fatty acids used in this study.

2.Please provide the characteristics of each short-chain fatty acid.

3.Please suggest the (different) underlying mechanism for each short-chain fatty acid.

Thank you very much.

Author Response

  1. Please list the types of short-chain fatty acids and the short-chain fatty acids used in this study.

AUTHORS: We appreciate your recommendation and the short-chain fatty acids employed were listed in our Results section.

  1. Provide the characteristics of each short-chain fatty acid.

AUTHORS: Complementary information of the short-chain fatty acids included in this systematic review were added to our Results section. Moreover, a brief description of the SCFA can be find in our Introduction.

  1. Please suggest the (different) underlying mechanism for each short-chain fatty acid.

AUTHORS: We accepted the suggestion of the reviewer. The different mechanisms by which short-chain fatty acids can operate on epithelial cells were suggested by the authors and further discussed in details in our Discussion section. Moreover, we refined our Introduction section, and more information concerning the possible underlying mechanisms of short-chain fatty acids were presented.

 Thank you very much.

AUTHORS: We appreciate your time and your worthy comments about our article.

Round 3

Reviewer 3 Report

Thank you very much for submitting your revised research to International Journal of Molecular Sciences.

It seems that all the queries were answered very well.

Thank you very much.